# Unsupervised Stain Decomposition via Inversion Regulation for Multiplex Immunohistochemistry Images

**Shahira Abousamra**[1]                                    SABOUSAMRA@CS.STONYBROOK.EDU
[1] *Stony Brook University, Department of Computer Science, USA*
**Danielle Fassler**[2]                          DANIELLE.FASSLER@STONYBROOKMEDICINE.EDU
[2] *Stony Brook University, Department of Pathology, USA*
**Jiachen Yao**[1]                                            JIACYAO@CS.STONYBROOK.EDU
**Rajarsi Gupta**[3]                             RAJARSI.GUPTA@STONYBROOKMEDICINE.EDU
[3] *Stony Brook University, Department of Biomedical Informatics, USA*
**Tahsin Kurc**[3]                                        TKURC@STONYBROOKMEDICINE.EDU
**Luisa Escobar-Hoyos**[2,4]                            LUISA.ESCOBAR-HOYOS@YALE.EDU
[4] *Yale University, Department of Therapeutic Radiology, USA*
**Dimitris Samaras**[1]                                      SAMARAS@CS.STONYBROOK.EDU
**Kenneth Shroyer**[2]                     KENNETH.SHROYER@STONYBROOKMEDICINE.EDU
**Joel Saltz**[3]                                    JOEL.SALTZ@STONYBROOKMEDICINE.EDU
**Chao Chen**[3]                                        CHAO.CHEN.1@STONYBROOK.EDU

**Editors:** Accepted for publication at MIDL 2023

## Abstract

Multiplex Immunohistochemistry (mIHC) is a cost-effective and accessible method for in situ labeling of multiple protein biomarkers in a tissue sample. By assigning a different stain to each biomarker, it allows the visualization of different types of cells within the tumor vicinity for downstream analysis. However, to detect different types of stains in a given mIHC image is a challenging problem, especially when the number of stains is high. Previous deep-learning-based methods mostly assume full supervision; yet the annotation can be costly. In this paper, we propose a novel unsupervised stain decomposition method to detect different stains simultaneously. Our method does not require any supervision, except for color samples of different stains. A main technical challenge is that the problem is underdetermined and can have multiple solutions. To conquer this issue, we propose a novel inversion regulation technique, which eliminates most undesirable solutions. On a 7-plexed IHC images dataset, the proposed method achieves high quality stain decomposition results without human annotation.

**Keywords:** Color deconvolution, Deep learning, Multiplex, Immunohistochemistry, Unsupervised.

## 1. Introduction

Multiplex Immunohistochemistry (mIHC) is a tissue imaging technique that allows simultaneous detection of multiple biomarkers in the same tissue section. Multiplex imaging has become an important tool in diagnostic pathology and cancer immunotherapy. It enables the study of cell composition, cellular function and cell-cell interactions, not to mention, allowing a more in depth comprehension and profiling of the tumor microenvironment (Ilié

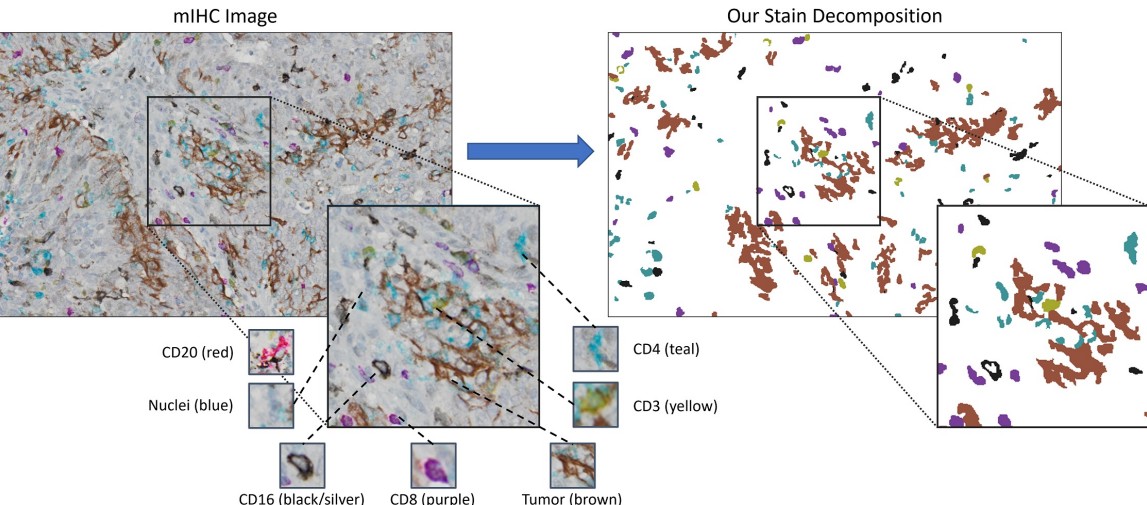

Figure 1: An mIHC image and the corresponding unsupervised stain decomposition. Samples from the different biomarkers are shown. The CD20 sample is from a different image since it is not present in this view. The yellow staining looks like olive yellow because of the interaction with the blue nuclei staining.

et al., 2018; Johnson et al., 2018; Remark et al., 2016; Tan et al., 2020). While there are various multiplex imaging technologies, brightfield mIHC is a cost effective and accessible method that can be captured in a single step with traditional brightfield light microscopy. Moreover, it does not require any special software or visualization tools, making biomarker research and profiling more accessible (Tan et al., 2020). MIHC offers simultaneous staining of 5+ biomarkers (Tan et al., 2020; Fassler et al., 2020; Hofman et al., 2019; Dixon et al., 2015). Unlike more expensive multiplex imaging techniques, e.g., multiplex immunofluorescence, mIHC does not offer separate channels for different biomarkers. It rather provides a single image with all the biomarkers visualized together, see Figure 1 (left). The simultaneous staining in mIHC images can offer insights into the tumor microenvironment that are not obtainable with singly stained IHC images but it still requires much manual effort to analyze. Visual inspection is both challenging and time consuming for pathologists, especially with increasing number of stains.

To analyze the images at a large scale, this process needs to be automated. The images need to be decomposed into individual biomarker maps before any spatial analysis or quantification can take place (Compare the left- and right-hand sides in Figure 1). To that end, digital analysis methods are essential to automate the stain unmixing, i.e., extracting masks corresponding to different biomarkers (Van Herck et al., 2021) to use them for tumor microenvironment analysis (Fassler et al., 2020).

Color deconvolution is the unmixing of stains into separate stain concentration maps. Several unsupervised color deconvolution methods (Ruifrok and Johnston, 2001; Vahadane et al., 2016; Macenko et al., 2009) have been successfully deployed for the analysis of up to two biomarkers, such as hematoxylin and eosin (H&E). Extending to a larger number of

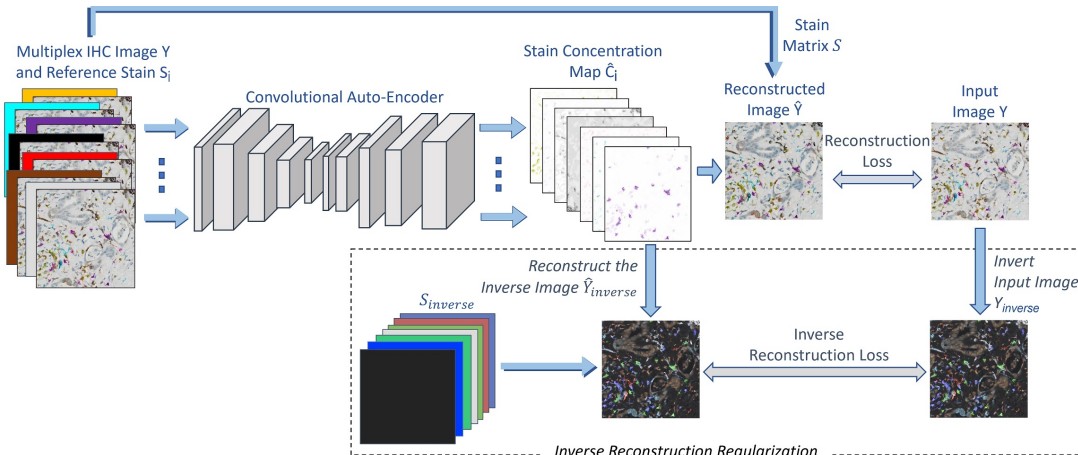

Figure 2: InverseAE architecture and training. The model's output is conditioned on the input mIHC image and the target reference stain. It is trained with a reconstruction loss and an inverse reconstruction loss.

stains involves solving an under-determined linear equation system, which can have many undesired solutions. To address this issue, in (Chen and Chefd'hotel, 2014; Chen and Srinivas, 2015), a lasso regression model is proposed to select a sparse set of stains at each pixel. The results, however, are not satisfying, mostly because these methods treat different pixels independently, and fail to consider the contextual information. In (Duggal et al., 2017) dual-stained images are implicitly deconvolved for downstream classification tasks, but the deconvolution channels are not guaranteed to accurately correspond to the different stains. Commercial tools such as HALO can unmix up to 4-colored chromogenic stains (Thommen et al., 2018). More recently, (Abousamra et al., 2020) proposed a weakly supervised deep learning approach for stain decomposition. Its performance depends on manual annotations of cell centers, which can be very time consuming.

In this work, we propose an unsupervised mIHC stain decomposition method that can scale to a larger number of stains. Our method only requires the reference stain colors. To properly leverage contextual information, we use a convolutional neural network autoencoder to generate the stain concentration maps. With an increased number of stains, the solution is not unique. Using the sparsity prior is not the best solution as it fails to consider the context. To this end, we propose a novel inverse-image-based regularization to help the network learn high quality stain concentration maps. We show how different cell types or biomarker segmentation maps can be directly inferred from the resulting concentration maps. On a 7-plex mIHC images dataset, we show the effectiveness of our proposed approach, achieving evaluation scores very close to the supervised method (Abousamra et al., 2020) ($\sim 94\%$ of the supervised f-score on average). Our code is available here: https://github.com/ShahiraAbousamra/MIHC-InverseAE

## 2. Related Work

Stain unmixing/decomposition is a classical problem that has been studied extensively on histological images, e.g., H&E. Ruifrok and Johnston (2001) were the first to use the optical density color representation to address the problem. The decomposition problem is transformed into solving a linear equation system. Different decomposition methods have been proposed (Macenko et al., 2009; Alsubaie et al., 2016, 2017; Vahadane et al., 2016; Hidalgo-Gavira et al., 2018; Pérez-Bueno et al., 2020).

Recent years have witnessed rapid advancement of staining technology; one may acquire simultaneous staining of more than 2 biomarkers in the same tissue section. See example in Figure 1. However, with mIHC allowing 5+ simultaneous stains (Tan et al., 2020), stain unmixing becomes more challenging and new methods are needed to handle a large number of stains. Chen and Chefd'hotel (2014) extended the work of (Ruifrok and Johnston, 2001) to support more than 2 biomarkers. With more than 2 biomarkers, the optical-density-based linear equation system is underdetermined. To address the issue, lasso regression was proposed. Nevertheless, directly enforcing a sparsity prior is not sufficient, especially for higher number of stains, mainly because the sparsity prior fails to consider contextual information of each pixel. (Chen and Srinivas, 2015) further adds constraints on the lasso regression to model co-staining. Abousamra et al. (2020) proposes to use weak supervision in the form of dot annotations that are transformed into superpixels (Achanta et al., 2012) to overcome the underdetermined nature of the problem. They train a convolutional autoencoder to estimate the concentration maps and show the effectiveness of the method in decomposing 6 simultaneous stained mIHC images. Their method, however, still requires a large amount of annotations, which can be prohibitive in practice. For completeness, we also refer to methods that directly incorporate color deconvolution into downstream tasks (Duggal et al., 2017; Lahiani et al., 2018), e.g., tissue segmentation.

## 3. Background

MIHC images are obtained by histologic detection of multiple protein biomarkers. In this section we describe the linear relationship of stain concentrations under Beer Lambert law. This is the basis for most stain unmixing methods including ours.

**Optical Density (OD).** Optical density measures the absorbance of light as it passes through a medium. The optical density at a pixel $i$ is the negative log of its normalized RGB values, $[r_i, g_i, b_i]^T \in [0, 1]^3$ :

$$\boldsymbol{y}_i = [-\log(r_i), -\log(g_i), -\log(b_i)]^T. \tag{1}$$

The stains and their concentrations are linearly related in the OD space. Assume a given set of $m$ stain vectors, whose OD vectors are represented by the matrix $\boldsymbol{S} = [\boldsymbol{s}_1, \boldsymbol{s}_2, \cdots, \boldsymbol{s}_m] \in \mathbb{R}^{3 \times m}$, where $\boldsymbol{s}_j \in \mathbb{R}^3$ is the OD vector of the $j^{th}$ stain. By Beer Lambert Law (BL) (Lambert, 1760), for any pixel $i$, the optical density $\boldsymbol{y}_i$ and its stain concentration vector $\boldsymbol{c}_i$ satisfy a linear relationship, i.e., $\boldsymbol{y}_i = \boldsymbol{S} \cdot \boldsymbol{c}_i$, where $\boldsymbol{c}_i \in \mathbb{R}^m$ denotes the concentrations of the different stains at pixel $i$. Generalizing to an image with $n$ pixels, we get:

$$\boldsymbol{Y} = [\boldsymbol{y}_1, \cdots, \boldsymbol{y}_n] = \boldsymbol{S} \cdot \boldsymbol{C} \tag{2}$$

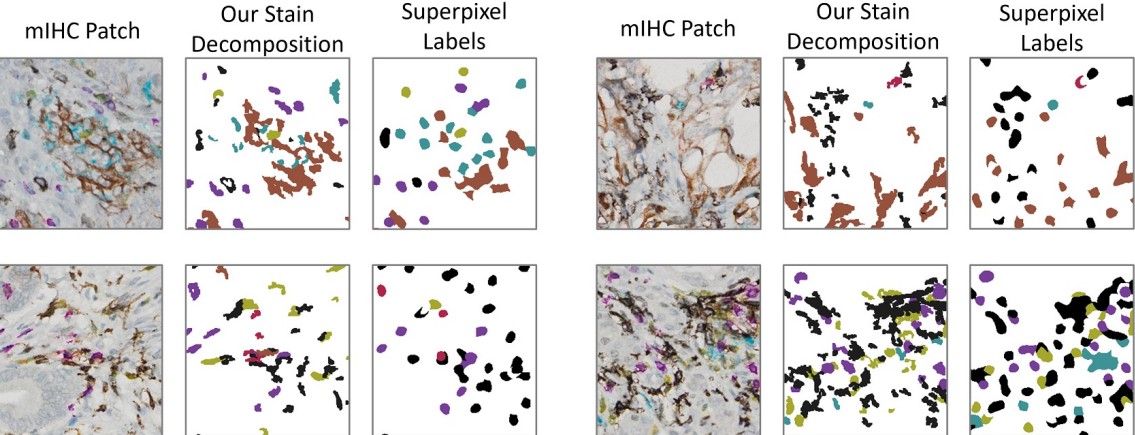

Figure 3: Sample mIHC patches, our corresponding unsupervised stain decomposition, and the superpixel labels derived from the dot annotations. The superpixel labels are obtained using the algorithm in (Achanta et al., 2012).

in which $\boldsymbol{Y} \in \mathbb{R}^{3 \times n}$ is the optical density matrix of all the pixels in the image and $\boldsymbol{C} = [\boldsymbol{c}_1, \cdots, \boldsymbol{c}_n] \in \mathbb{R}^{m \times n}$ is the corresponding stains' concentrations matrix.

**Solving the Linear Equation.** The stain mixing was first modeled by (Ruifrok and Johnston, 2001) using Beer Lambert Law. Given the reference stains matrix $\boldsymbol{S}$, they proposed to recover the stain concentrations by solving the linear equation system, Equation (2). In practice, stain color vectors are obtained by sampling representative pixels of each stain and the background tissue. The background color vector prevents the background tissue from getting erroneously computed as a mixture of other stains. For higher than 2-plex staining i.e., when the stain matrix $S$ consists of $m > 3$ stain vectors (2 stains and the background), the linear equation system is underdetermined and does not have a unique solution.

## 4. Method: InverseAE for Unsupervised Stain Decomposition

In this work we propose *InverseAE*, an unsupervised deep-learning stain decomposition method. We assume that at each pixel there is a dominant stain which should be assigned to that pixel in the final stain segmentation maps. Hence, our proposed method is designed so that it results in meaningful stain concentration maps from which the dominant stain/biomarker at each pixel can be inferred directly. It only requires the set of multiplex stained images and the set of reference stains color vectors.

We use an autoencoder to generate the estimated stain concentration maps $\widehat{\boldsymbol{C}}$ for an input image $\boldsymbol{Y}$, based on the given stain vectors $\boldsymbol{S}$. The autoencoder is trained with the reconstruction loss: $\mathcal{L}_{recon}(\widehat{\boldsymbol{C}}) = MSE(\boldsymbol{Y}, \boldsymbol{S} \cdot \widehat{\boldsymbol{C}})$ that is aimed to solve Equation (2). It compares the image $\boldsymbol{Y}$ with the reconstructed image from the estimated concentration maps $\boldsymbol{S} \cdot \widehat{\boldsymbol{C}}$ using the mean squared error. When the number of stain vectors $m > 3$, Equation (2) becomes an underdetermined system of equations with possibly infinite number of solutions. We observe that when we train the autoencoder with only the reconstruction loss, the model

may (and oftentimes tends to) converge to suboptimal solutions because of the under-determined nature of the optical density based linear equation system, Equation (2). To that end, we propose the inverse reconstruction regularizer to impose additional constraints.

### 4.1. Inversion Regulation

The inverse reconstruction regularizer implements a constrained training mechanism by promoting a solution –estimated concentration maps $\widehat{C}$– that satisfies both the image and its inverse. We use the term *inverse of an image* or *inverted image* to refer to the image formed from the pixels' complement color vectors, i.e. for an arbitrary pixel with normalized RGB color vector $[r,\ g,\ b]^T$, its *inverted color vector* is $[1-r,\ 1-g,\ 1-b]^T$, and its *inverted optical density* is $\boldsymbol{y}_{inverse} = [-\log(1-r), -\log(1-g), -\log(1-b)]^T$. Similarly, $\boldsymbol{S}_{inverse}$ is the inverted optical density for the stain matrix $\boldsymbol{S}$. To regularize the solution of Equation (2), we add the inverted equations into the system.

$$\text{For each pixel:} \quad \boldsymbol{y}_{inverse} = \boldsymbol{S}_{inverse} \cdot \boldsymbol{c}, \tag{3}$$

$$\text{For the image:} \quad \boldsymbol{Y}_{inverse} = \boldsymbol{S}_{inverse} \cdot \boldsymbol{C}, \tag{4}$$

Therefore, the inverse reconstruction regularization loss is defined as:

$$\mathcal{L}_{inverse}(\widehat{\boldsymbol{C}}) = MSE(\boldsymbol{Y}_{inverse}, \boldsymbol{S}_{inverse} \cdot \widehat{\boldsymbol{C}}). \tag{5}$$

And the overall loss function becomes:

$$\mathcal{L}_{unsup}(\widehat{\boldsymbol{C}}) = \lambda_1 \mathcal{L}_{recon}(\widehat{\boldsymbol{C}}) + \lambda_2 \mathcal{L}_{inverse}(\widehat{\boldsymbol{C}}) \tag{6}$$

Where $\lambda_1$ and $\lambda_2$ are constant weight values. In other words, the model is trained to generate concentration maps $\widehat{C}$ that when combined with the stain matrix $\boldsymbol{S}$, reconstructs the input image $\boldsymbol{Y}$, and meanwhile, when combined with the inverted stain matrix $\boldsymbol{S}_{inverse}$, reconstructs the inverted image $\boldsymbol{Y}_{inverse}$. We show in the experiments how this regularization pushes the prediction towards *good solutions*, that is, at any location, the stain corresponding to the observed dominant color gets the highest concentration. This allows us to infer high quality stain segmentation maps and achieve evaluation scores close to the supervised solution. We next describe the intuition behind the inversion regulation.

This choice of inverse as a regulator is not random. A suitable regulator requires 2 **properties**: **(I)** a solution exists that satisfies both the original image and the regulator, **(II)** the regulator adds constraints that limit the solution space by eliminating suboptimal solutions and promoting desired solutions. These properties prompt our choice of the inverse. First, there is a special relationship between a color and its complement (or inverse). When an object absorbs light of a particular color, we perceive the object as having the complementary color, and vice versa. Indeed, the relationship can be stated quantitatively. A color and its inverse share the same saturation in the HSL color space (see Appendix B for details). Saturation measures the departure of a hue from white (Agoston, 2005) and thus correlates with the amount of pigment present, i.e., its concentration. These observations suggest that a color and its inverse can share a same concentration vector, with regard to the original stains and the inverse stains, respectively. In other words, the same $\boldsymbol{c}$ can satisfy both $\boldsymbol{y} = \boldsymbol{S} \cdot \boldsymbol{c}$ and $\boldsymbol{y}_{inverse} = \boldsymbol{S}_{inverse} \cdot \boldsymbol{c}$. This is the desired *property I*.

Table 1: F-score of stain segmentations with respect to dot annotations on 96 patches from 42 cases. Shown in bold and underlined are the best and second best F-scores in each column, respectively.

| Method | CD3 | CD4 | CD8 | CD16 | CD20 | Tumor | Mean |
|---|---|---|---|---|---|---|---|
| PseudoInv | 0.158 | 0.208 | 0.441 | 0.267 | 0.265 | 0.403 | 0.290 |
| Nearest Neighbor | 0.223 | 0.168 | 0.734 | 0.277 | 0.106 | 0.357 | 0.311 |
| LassoReg | 0.456 | 0.501 | 0.801 | 0.050 | 0.223 | 0.389 | 0.404 |
| UNet (supervised) | 0.496 | 0.496 | **0.835** | 0.699 | 0.144 | 0.405 | 0.513 |
| ColorAE (supervised) | 0.639 | **0.585** | 0.829 | **0.728** | **0.353** | **0.573** | **0.618** |
| InverseAE | **0.644** | 0.530 | 0.813 | 0.632 | 0.327 | 0.556 | 0.584 |

Table 2: Evaluation of concentration maps using SSIM score on single stain images.

| Method | CD3 | CD4 | CD8 | CD16 | CD20 | Tumor | Mean |
|---|---|---|---|---|---|---|---|
| ColorAE | 0.893 | **0.886** | **0.897** | 0.657 | 0.876 | **0.834** | 0.841 |
| InverseAE | **0.935** | 0.881 | 0.879 | **0.682** | **0.881** | 0.800 | **0.843** |

Second, in the HSL color space, a color and its inverse have contrasting luminosity, i.e. when the luminosity of a color is $L$, the luminosity of its inverse is $1 - L$, see Appendix B. We can define the absorbance power of a color as the average of the minimum and maximum of its red, green, and blue light absorbance. Then, by definition, its luminosity is inversely correlated with its absorbance power. As a result, *since a color and its inverse have contrasting luminosity, they also have contrasting absorbance powers.* This property allows inversion regulation to eliminate a large portion of the undesired solution space and promote good sparse solutions (*property II*). We discuss this in more detail in Appendix C.

We have shown how the inversion regulation brings special properties that are essential to converge to good solutions and hence, get high accuracy stain segmentation maps. Using the inverse reconstruction loss on a mIHC dataset with $m = 8$ stain vectors, we observe that it pushes the prediction towards the desired solution, that is, at any location, the stain corresponding to the observed dominant color gets the highest concentration. Figure 2 shows an overview of the model architecture and training.

## 5. Experiments

We performed experiments with 7-plex mIHC stained pancreatic cancer tissue. The 7 biomarkers include immune cell biomarkers: CD3 (yellow), CD4 (teal), CD8 (purple), CD16 (black), CD20 (red), a tumor biomarker (brown), in addition to hematoxylin as a nuclear counterstain (blue). The training dataset consisted of 240 patches of size $400 \times 400$ at $20x$ divided into training and validation. See Appendix A for more details on the training and inference, including how we obtain stain segmentations from the estimated concentration maps. Next, we present our experiments for evaluating the predicted segmentation maps and for evaluating the quality of the raw concentration maps. An ablation study of the effect of different components on the model is in Appendix D.1. Additional quantitative

and qualitative results are in Appendix D. A discussion of the results, limitations, future work, and impact is presented in Appendix E.

**Evaluation with Dot Annotations.** We tested the model on 96 patches of size $1200 \times 1920$ from 42 cases. These test patches have manual dot annotations for each biomarker where each dot marks the approximate center of a cell. The dot annotations are expanded to superpixels (Achanta et al., 2012) for visualization of the ground truth labels in Figure 3 and Figure 6. We compute the F-score for the predicted semantic segmentation maps for each biomarker, based on whether or not the generated segments intersect with the dot annotations of the corresponding biomarker. Details of F-score computation are in Appendix A. Table 1 shows the F-score results. We compare our method *InverseAE* to the supervised models: *ColorAE* and *UNet* (Ronneberger et al., 2015), and to the unsupervised methods: *LassoReg* (Chen and Chefd'hotel, 2014), *PseudoInv.* which is an extension of (Ruifrok and Johnston, 2001) to support a stain matrix with $> 3$ stains using matrix pseudo inverse, and *Nearest Neighbor*, which basically assigns the stain with the closest Euclidean distance to the pixel color value. See Appendix A for a description of the baselines. From Table 1, *InverseAE* outperforms all other unsupervised methods, coming at close second after *ColorAE*. Figure 3 shows sample results from *InverseAE*, and in Appendix D.6 we show sample results comparing against the baselines. A further discussion of the results is presented in Appendix E.

**Evaluation with Ground Truth Concentration Maps.** We assessed the quality of the concentration maps of the trained *InverseAE* and *ColorAE* models compared to 2-plex SOTA stain decomposition method (Vahadane et al., 2016) on a set of IHC 2-plex images using structure similarity index (SSIM). The 2-plex images were stained with one of the immune/tumor biomarkers plus hematoxylin. The results are in Table 2. Interestingly, the SSIM scores for *InverseAE* are often better than those for *ColorAE* showing that it better captures the continuous values of the concentration maps. Additional evaluations of the quality of the generated concentration maps are in Section D.3

## 6. Conclusion

While brightfield mIHC is a cost effective and accessible multiplex imaging method, it does not offer separate channels for different biomarkers. Furthermore, existing unsupervised stain decomposition methods do not scale well with the increase in number of multiplexed stains. This makes brightfield mIHC impractical compared to more expensive multiplex imaging methods. In this work, we have tackled the problem of stain decomposition in mIHC images and shown the efficacy of our proposed approach on mIHC images with 7 simultaneous stains. Our unsupervised approach, '*InverseAE*' , allows pathologists to get accurate results while relieving them from the burden of annotation. We propose a novel training regularizer for stain decomposition, '*inversion regulation*', to create additional constraints during training that limit the solution space. Hence *InverseAE* learns to capture the correct cell types, and the resulting stain concentration maps are representative of the underlying staining. Consequently, *InverseAE* enables high quality stain segmentation maps which are essential for downstream analysis.

## 7. Acknowledgements

We would like to acknowledge the Stony Brook Cancer Center Biorepository for providing de-identified tissue specimens and The Research Foundation for The State University of New York at Stony Brook. This work was support by NSF grants IIS-2123920, IIS-2212046, and CCF-2144901, the National Institutes of Health (NIH) and National Cancer Institute (NCI) grants UH3-CA22502103, 3U24CA215109-02, 1UG3CA225021-01, and 5R01CA253368, as well as generous private support from Bob Beals and Betsy Barton.

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

## Appendix A. Training and Implementation Details

**Training Details.** InverseAE has a similar backbone network architecture to ColorAE. They are both shallow autoencoders with around 200K trainable parameters. The main difference is that InverseAE is conditional on the input target stain. The constant weight values $\lambda_1$ and $\lambda_2$ in Equation 6 are both set to 1 in our experiments. The training dataset consists of 60 annotated patches from 6 whole slide images. The size of the patches is $1200 \times 1920$ at 40x. 40 Patches from 4 cases were used for training and the remaining 20 patches from the other 2 cases were used for validation. For training, the images are resized to 20x and cut up into $400 \times 400$ overlapping patches, resulting in a total of 240 labeled training patches.

**Models Compared.** We train our proposed method *InverseAE* on the 240 patches, without annotation. We compare to the unsupervised approaches:
(1) *LassoReg*: the unsupervised lasso regression method described in (Chen and Chefd'hotel, 2014) which is designed to work on images with 4+ multiplex staining. We used L1 norm weight $\lambda = 10^{-3}$.
(2) *PseudoInv*: an extension of (Ruifrok and Johnston, 2001) for color deconvolution that solves the equation $\boldsymbol{C} = \boldsymbol{S}^{-1} \times \boldsymbol{Y}$ for more than 3 stain vectors using pseudo inverse of $\boldsymbol{S}$.
(3) *Nearest Neighbor* baseline which basically assigns the stain with the closest Euclidean distance to the pixel color value.
Additionally, we use the dot annotations to train with weak supervision:
(4) *ColorAE*: the weakly supervised method in (Abousamra et al., 2020).
(5) *UNet*: a supervised deep learning model based on UNet model architecture (Ronneberger et al., 2015). Different from previous methods, UNet's output is not stain concentration maps but rather likelihood maps. For each pixel, the output is the probability that the pixel belongs to each stain class. Superpixel annotation maps (Achanta et al., 2012) derived from the dot labels are used as the ground truth labels.

**Stain Segmentation Maps Inference.** The ultimate goal is to get a semantic segmentation map of the stains. Such a map would reflect the presence of different biomarkers or cell types to utilize in downstream analysis. The resulting concentration from our unsupervised solution can be used directly to do just that without further annotation or training. Each stain has a minimum concentration value below which the stain can be excluded from the competition for the dominating stain. During inference, each stain concentration map is thresholded using hysteresis thresholding and then an argmax function is applied to select the stain with the maximal concentration at each location. The resulting segmentation map has a single stain prediction at each pixel, reflecting the predicted dominant stain at each location. The hysteresis thresholding ensures that for each stain we only get regions where there is high confidence in the presence of that stain and then expand it to cover the larger cell area. The threshold parameters are fixed for each stain and are empirically selected on a small held out set. See sample results in Figure 3.

**F-score Computation.** We computed the F-score for each biomarker/cell type, as follows: a true positive is a predicted stain segment that intersects with an annotation dot, a false positive is a segment that doesn't intersect with any annotation dot, a false negative is an annotation dot that does not intersect with any predicted segment.

## Appendix B. RGB to HSL Conversion.

We include here the formulas for obtaining saturation and lightness from RGB for their relevance. The complete conversion algorithm including hue can be found in (Agoston, 2005). To convert from a normalized RGB color $(r, g, b)$ to HSL space, we first define the variables $M$ and $m$ that represent the maximum and minimum of $(r, g, b)$, respectively :

$$M = max(r, g, b) \tag{7}$$

$$m = min(r, g, b) \tag{8}$$

The formula for luminosity L:

$$L = \frac{1}{2}(M + m) \tag{9}$$

The formula for saturation $\mathcal{S}$:

$$\mathcal{S} = \begin{cases} 0 & \text{if } L = 0 \text{ or } L = 1. \\ \frac{M-m}{1-|2L-1|} & \text{otherwise.} \end{cases} \tag{10}$$

For the inverted color $(r', g', b') = (1 - r, 1 - g, 1 - b)$, the formulas become:

$$M' = 1 - m \tag{11}$$

$$m' = 1 - M \tag{12}$$

$$L' = 1 - L \tag{13}$$

$$\mathcal{S}' = \begin{cases} 0 & \text{if } L' = 0 \text{ or } L' = 1. \\ \frac{M'-m'}{1-|2L'-1|} & \text{otherwise.} \end{cases} = \begin{cases} 0 & \text{if } L = 0 \text{ or } L = 1. \\ \frac{M-m}{1-|2L-1|} & \text{otherwise.} \end{cases} = \mathcal{S} \tag{14}$$

## Appendix C. Inversion Regulation Eliminates Many Undesired Solutions

We assume that at each pixel there is a dominant stain which should be assigned to that pixel in the final stain segmentation maps. If we consider the luminosity component in the HSL color space, when the luminosity of a color is $L$, the luminosity of its inverse is $1 - L$, see Appendix B. Now, if we define the absorbance power of a color as the average of the minimum and maximum of its red, green, and blue light absorbance, then by definition (Agoston, 2005), its luminosity is inversely correlated with it's absorbance power. As a result, since a color and its inverse have opposite luminosity, they also have opposite absorbance powers. We show here how this property allows us to eliminate a large part of the undesired solution space by employing the inverse regulation.

BL allows a color to be made up of multiple colors added together weighted by their concentrations. Consequently, in the OD space, a pixel's color can be composed by the addition of multiple stains -other than the true stain present at that pixel- as long as their red, green, and blue OD (absorbance) values sum up to the color's red, green, and blue OD (absorbance) values, respectively. This is an undesired solution. However, this solution will not hold in the inverse space as long as no stain is a function of another by a multiplicative factor. We can see this by a simple example: A stain vector representing white

or the background tissue will have very low absorbance, for instance $(250, 250, 250)_{RGB} \implies (0.01, 0.01, 0.01)_{OD}$. By BL, it can be added to any color without modifying the color's OD. Nevertheless, in the inverse space, this does not hold because the complement color has high absorbance (OD): $(5, 5, 5)_{inv-RGB} \implies (1.7, 1.7, 1.7)_{inv-OD}$. As a result, it will modify the OD of any pixel it is added to its mixture and thus it cannot be added blindly. Another example, consider the 3 stain vectors:

$SA_{RGB} = (250, 100, 100) \implies SA_{OD} \simeq (0.0, 0.4, 0.4)$
$SB_{RGB} = (250, 250, 100) \implies SB_{OD} \simeq (0.0, 0.0, 0.4)$
$SC_{RGB} = (250, 100, 250) \implies SC_{OD} \simeq (0.0, 0.4, 0.0)$
By BL, $SA$ can be expressed as a linear combination of $SB$ and $SC$.

$$SA_{OD} = SB_{OD} + SC_{OD} \tag{15}$$

Now if we examine the same stains in the inverse space:
$SA_{inv-RGB} = (5, 155, 155) \implies SA_{inv-OD} \simeq (1.7, 0.2, 0.2)$
$SB_{inv-RGB} = (5, 5, 155) \implies SB_{inv-OD} \simeq (1.7, 1.7, 0.2)$
$SC_{inv-RGB} = (5, 155, 5) \implies SC_{inv-OD} \simeq (1.7, 0.2, 1.7)$
Equation 15 does not hold in the inverse space.

$$SA_{inv-OD} \neq SB_{inv-OD} + SC_{inv-OD} \tag{16}$$

This illustrates how the inverse regulation helps to avoid many such undesired solutions and promote good solutions.

## Appendix D. Additional Experiments

### D.1. Ablation Study: Model Design Choices

Here we present ablation studies on the design choices for our unsupervised model *InverseAE*. In Table 3 we present the F-score results on the 96 annotated patches using variations from our proposed method. First, the results show the importance of the inverse reconstruction loss. Without it (rows 1-3), the predicted maps are not representative of the stains visually present in the mIHC image. This is clear from the very low F-scores. Second, replacing the inverse reconstruction loss with a L1 sparsity constraint (row 2) is not sufficient to get representative concentration maps. We notice sparsity in the prediction, nevertheless the prediction is not correlated with the observed stains. We find the dominating stains in the prediction are mostly CD3's yellow and CD8's purple, while the tumor (brown) and the CD16 (black) biomarkers are missing, as evidenced by their zero F-score in the table. Third, instead of the inverse regulation loss, we use the inverted images and stains as data augmentation (row 3). We observe that we get slightly improved performance compared to the L1 sparsity constraint (row 2), however it also suffers from missing biomarkers in the predicted concentration maps. Finally, without the conditional autoencoder (row 4) i.e., predicting all the stains at once without target stains conditional input, the scores are lower and the model does not generalize well. The conditional input prevents the model from overfitting and allows the model to better capture the target stains in the image.

Table 3: Ablation study on unsupervised model design InverseAE, evaluated with F-score of stain segmentations with respect to dot annotations on 96 patches from 42 cases.

| Inverse Reg. | Conditional AE | $L1$ Sparsity | Inv. Aug. | CD3 | CD4 | CD8 | CD16 | CD20 | Tumor | Mean |
|---|---|---|---|---|---|---|---|---|---|---|
| | ✓ | | | 0.235 | 0.307 | 0.409 | 0.121 | 0.326 | 0.147 | 0.257 |
| | ✓ | ✓ | | 0.215 | 0.228 | 0.337 | 0.000 | 0.300 | 0.000 | 0.180 |
| | ✓ | | ✓ | 0.407 | 0.411 | 0.716 | 0.000 | 0.117 | 0.000 | 0.275 |
| ✓ | | | | 0.482 | 0.501 | 0.753 | 0.324 | 0.274 | 0.220 | 0.426 |
| ✓ | ✓ | | | **0.644** | **0.530** | **0.813** | **0.632** | **0.327** | **0.556** | **0.584** |

Table 4: Ablation study measuring the impact of $\mathcal{L}_{inverse}$ in Equation 6, by fixing $\lambda_1 = 1$ and modifying $\lambda_2$.

| $\lambda_2$ | CD3 | CD4 | CD8 | CD16 | CD20 | Tumor | Mean |
|---|---|---|---|---|---|---|---|
| 0.0 | 0.235 | 0.307 | 0.409 | 0.121 | 0.326 | 0.147 | 0.257 |
| 0.25 | 0.545 | 0.589 | 0.803 | 0.670 | 0.224 | **0.620** | 0.575 |
| 0.5 | 0.531 | 0.476 | 0.791 | 0.681 | 0.273 | 0.613 | 0.561 |
| 0.75 | 0.580 | 0.573 | 0.790 | 0.698 | 0.268 | 0.581 | 0.582 |
| 1.0 | **0.644** | 0.530 | **0.813** | 0.632 | 0.327 | 0.556 | 0.584 |
| 1.25 | 0.629 | **0.602** | 0.781 | **0.709** | 0.320 | 0.475 | 0.586 |
| 1.5 | 0.584 | 0.584 | 0.809 | 0.694 | 0.327 | 0.559 | **0.593** |
| 2.0 | 0.625 | 0.571 | 0.791 | 0.634 | **0.333** | 0.532 | 0.581 |

**D.2. Ablation Study: Impact of $\lambda_2$**

In Table 4, we analyze the impact of $\lambda_2$ (the weight of $\mathcal{L}_{inverse}$ in Equation 6). We fix $\lambda_1 = 1$ and train with different values of $\lambda_2$. We observe that adding $\mathcal{L}_{inverse}$, even with a small weight ($\lambda_2 = 0.25$), can largely boost the performance compared to training without $\mathcal{L}_{inverse}$. This shows the power of the proposed Inverse Regulation Loss. The overall performance is quite robust/stable with regard to different $\lambda_2$ values. We find it interesting that as we increase $\lambda_2$ from 0.25 to 2, the model has a slightly increased performance on some immune biomarkers (e.g., CD3 and CD20), whereas its performance decreases on tumor. We conjecture that a heavier inverse regulation loss is making more detection on immune biomarkers by decreasing its detection preference on tumors, which is much more frequent in tumor microenvironment.

**D.3. Quality of Stain Concentration Maps**

Previously we evaluated the structural similarity (SSIM) of the stain concentration maps generated by InverseAE on 2-plex IHC images, where InverseAE model was trained on the mIHC dataset. The SSIM scores were reported in Table 2. This test set consisted of 20 patches of size $1000 \times 1000$ pixels at 20x magnification. The patches were sampled from 2 WSIs per stain, that is 120 patches sampled from 12 WSIs in total. Here, we include additional metrics: peak signal-to-noise ratio (PSNR) (psn, 2022) and learned perceptual image patch similarity (LPIPS) (Zhang et al., 2018). The PSNR is the ratio between the

Table 5: Evaluation of InverseAE concentration maps on single stain images with PSNR and LPIPS scores.

| Method | CD3 | CD4 | CD8 | CD16 | CD20 | Tumor | Mean |
|---|---|---|---|---|---|---|---|
| SSIM↑ | 0.935 | 0.881 | 0.879 | 0.682 | 0.881 | 0.800 | 0.843 |
| PSNR↑ | 32.83 | 25.78 | 27.15 | 20.97 | 30.83 | 24.47 | 27.01 |
| LPIPS↓ | 0.28 | 0.34 | 0.38 | 0.44 | 0.48 | 0.50 | 0.40 |

Table 6: Evaluation of InverseAE mIHC image reconstruction from generated concentration maps using SSIM, PSNR, LPIPS, and MSE scores.

| Method | SSIM↑ | PSNR↑ | LPIPS↓ | MSE↓ |
|---|---|---|---|---|
| InverseAE | 0.88 | 28.56 | 0.20 | 101.75 |

maximum possible power of a signal and the power of corrupting noise that affects the fidelity of its representation. It is expressed as a logarithmic quantity in decibels using the formula:

$$PSNR = 10 \log_{10} \frac{MAX_I^2}{MSE} \qquad (17)$$

where $MAX_I$ is the maximum possible pixel value of the image and MSE is the mean squared error. A higher PSNR indicates lower noise. LPIPS, on the other hand, computes the similarity between the activations of two image patches for some pretrained network. This metric has been shown to match human perception well. A low LPIPS score means that the image patches are perceptually similar. We compute LPIPS using a pretrained VGG network. The PSNR and LPIPS scores are reported in Table 5. The samples in Figure 4 show that the concentration maps generated by InverseAE largely match those generated by (Vahadane et al., 2016), except for regions with very light staining as in the case in the tumor brown stain example.

### D.4. Quality of mIHC Image Reconstruction

Here we evaluate the quality of the images reconstructed from InverseAE generated stain concentration maps. We use the metrics SSIM, PSNR, LPIPS, and MSE, as described in Section D.3. The scores are in Table 6. Figure 5 shows sample mIHC patches and the corresponding reconstruction from InverseAE stain concentration maps.

### D.5. Results over multiple iterations

In Table 7, we present the mean and standard deviation of InverseAE scores over 5 iterations. For each iteration we train on patches from $\frac{2}{3}$ of the whole slides for training and use patches from the remaining $\frac{1}{3}$ of the whole slides for validation. We evaluate all 5 models on the same test set used in Table 1. The results are fairly close to the single iteration results presented in Table 1 and the standard deviation is very small.

Table 7: Results from training InverseAE over multiple splits, Section D.5. For each category we report the mean and standard deviation of the F-score across 5 splits.

| Method | CD3 | CD4 | CD8 | CD16 | CD20 | Tumor | Mean |
|---|---|---|---|---|---|---|---|
| InverseAE | $0.602 \pm 0.04$ | $0.512 \pm 0.02$ | $0.809 \pm 0.00$ | $0.671 \pm 0.03$ | $0.297 \pm 0.02$ | $0.572 \pm 0.03$ | $0.577 \pm 0.01$ |

## D.6. Qualitative Results

In Figure 6, we expand the qualitative results to compare to the baseline methods. We observe nearest neighbor method is very sensitive to variations in the staining, resulting in various mis-classifications. LassoReg overestimates tumor (brown) and CD16 (black) regions and often mixes them up or mistakes the background tissue for them. Pseudo Inverse mistakes the background tissue for CD16 (black). It is worth noting that the mixup between the background and CD16 (black) is mainly due to the fact that the 2 colors have very close hues and mainly differ in luminosity. On the other hand, InverseAE and ColorAE most resemble the superpixel (Achanta et al., 2012) labels derived from the dot annotations.

## Appendix E. Discussion

We observe a variation in F-scores across different classes. More specifically, CD20 gets a very low score. This is caused by the very low distribution of CD20 across the dataset. The model sees less CD20 and thus tends to make more mistakes. Furthermore, with the small population in the test set, few errors are penalized heavily by the F-score. CD4 also suffers from a low training population compared to other classes, however not as low as CD20. Another common source of errors is the tumor staining (brown color) that is sometimes so dark that it appears as black and is thus predicted as CD16 (black stain) instead.

One limitation of our approach is that we do not model co-staining. We assume that there is a single dominant stain at each pixel and that it is the assigned label for that pixel. In the future, we plan to research methods to model co-staining. Moreover, we plan to integrate ColorAE (supervised) and InverseAE (unsupervised) to achieve scalable and high quality annotations of IHC images.

The real clinical benefit of our method lies in making multiplex IHC image analysis practical and accurate at the same time. Multiplex imaging is mostly used to study the tumor microenvironment by quantifying the distribution of different biomarkers and relating the quantification to cancer diagnosis and prognosis. Our method provides an efficient annotation solution for these IHC images, making the quantification much more scalable. This will potentially lead to more advanced diagnosis and treatment planning tools.

Other applications that might benefit from stain decomposition are those tasks that rely heavily on color and can benefit from color separation. One such application that comes to mind is aerial imaging, where color is the most prominent feature.

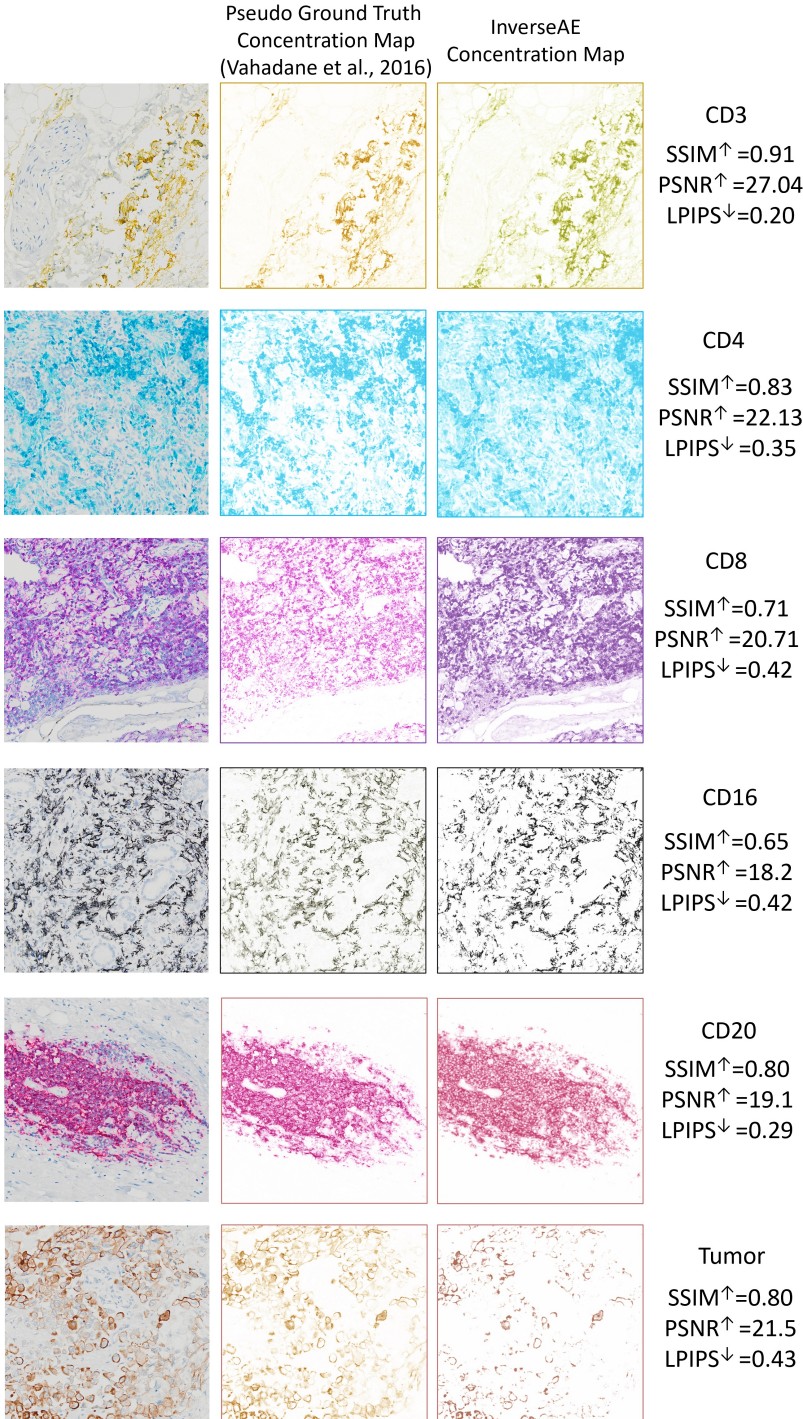

Figure 4: Sample IHC patches, the corresponding pseudo ground truth stain concentration maps obtained by (Vahadane et al., 2016), and InverseAE generated stain concentration maps. For each sample, the SSIM, PSNR, and LPIPS metrics are shown on the right.

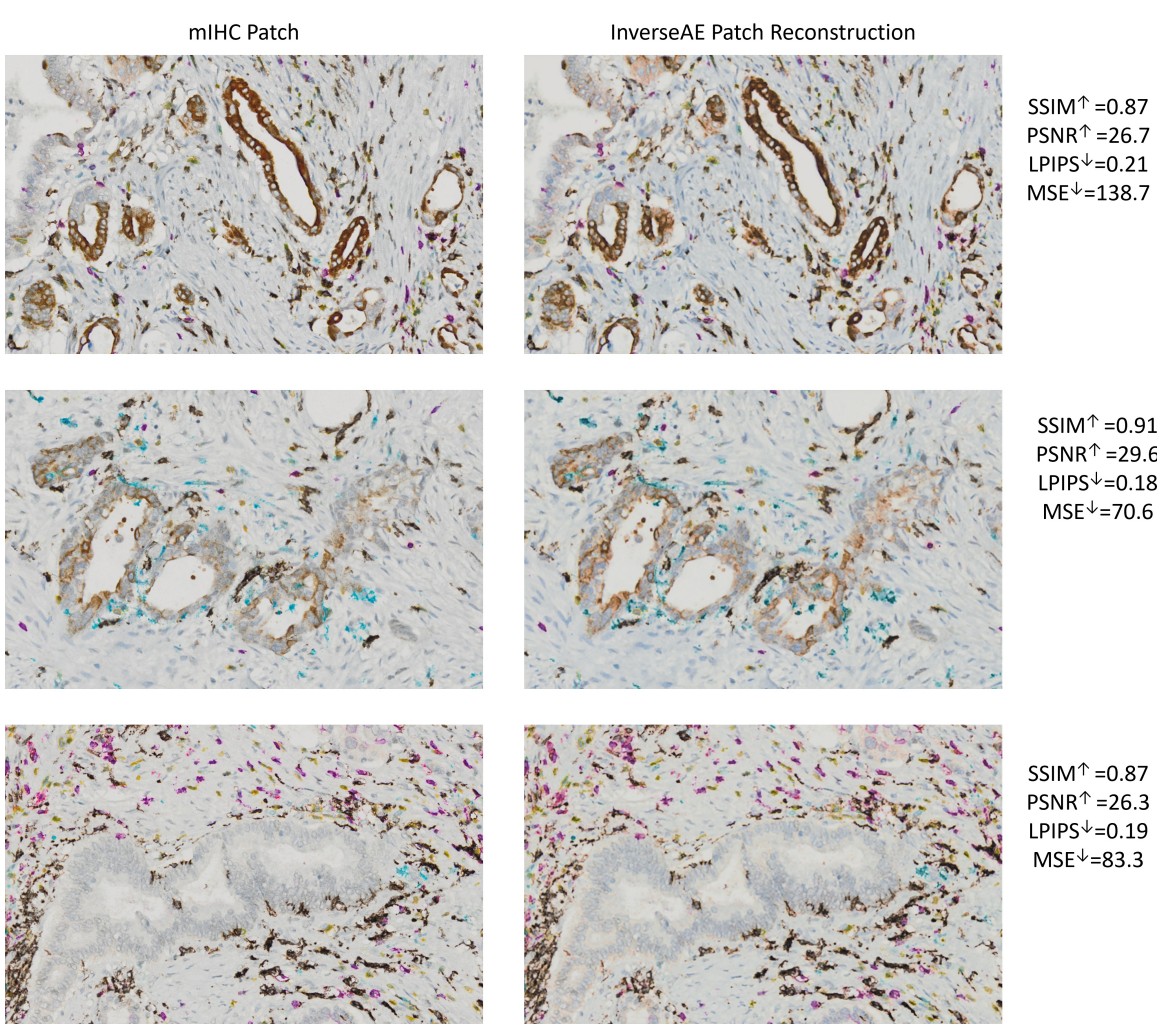

Figure 5: Sample mIHC patches reconstruction from InverseAE stain concentration maps.

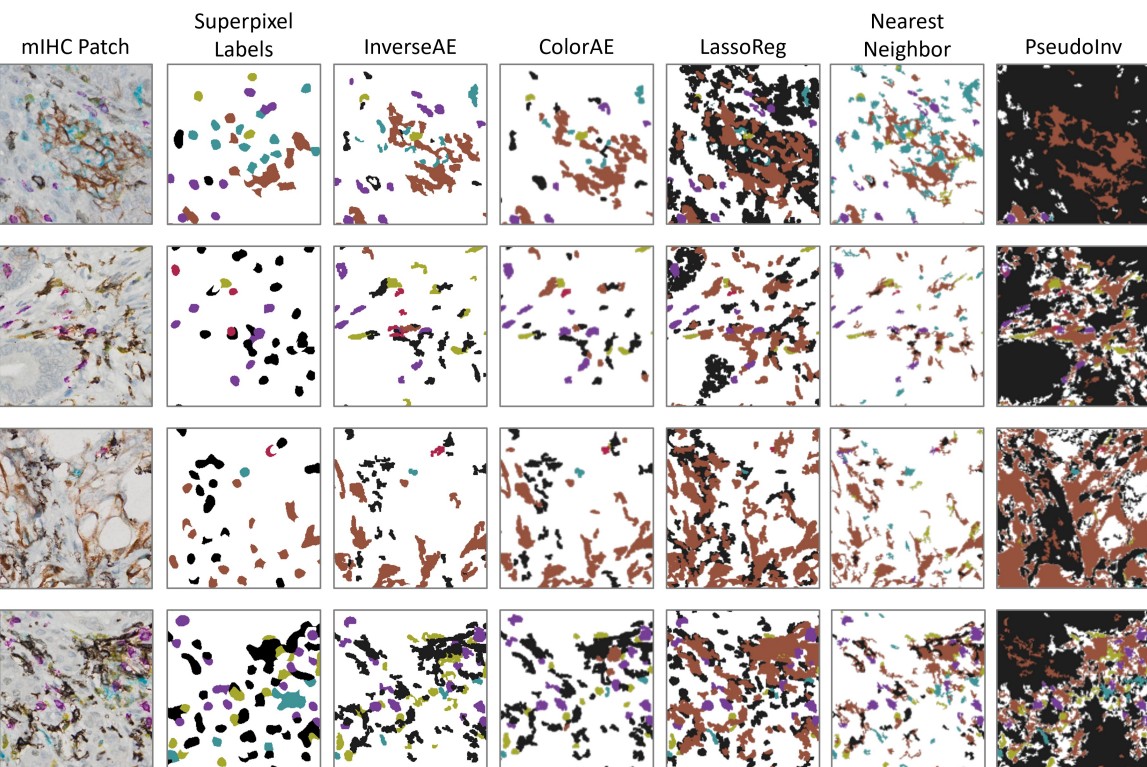

Figure 6: Sample stain segmentation results comparing InverseAE against the superpixel (Achanta et al., 2012) labels derived from the dot annotations, ColorAE, lassoReg, pseudo inverse, and nearest neighbor methods.

