# OpenReview forum: "Unsupervised Stain Decomposition via Inversion Regulation for Multiplex Immunohistochemistry Images"
_MIDL.io/2023/Conference — MIDL 2023 Oral_

### Official Review · Reviewer_bQMX · 2023-02-03

**Confidence:** 4
**Preliminary Rating:** 4
**Recommendation:** Poster

**Summary:**

This paper proposes a method for unsupervised stain decomposition to detect different stains from mIHC images.

The method is based on a CNN autoencoder (with the mIHC and reference stains as inputs) Stain concentration maps are estimated at the end of the decoder, prior to the reconstructed mIHC image calculated using the stain concentration map and the optical density matrix.
An inverse-image-based regularization is added to converge to good solutions.
The model is trained and validated using patches from 6 WSIs, and tested (for the dot-annotation evaluation) on patches from 42 WSIs.

**Strengths:**

The method is simple and the proposed inverse-image-based regularization method is well motivated and described.
The results are good, and compare well with those of supervised methods.
The experiments seem sufficient to show the benefit of the method.

**Weaknesses:**

I do not see major drawbacks. It is not a revolutionary method, but I think it is sufficiently good and interesting for the MIDL community for publication. Minor drawbacks and comments are listed below.

**Deanonymize Review:**

no

**Detailed Comments:**

Is it easy to obtain these reference stains color vectors? Maybe briefly comment.

In Figure 3, the superpixels are obtained with (Achanta et al., 2012) ? Mention in the caption.

“each stain concentration map is thresholded and then an argmax function is applied to select the stain with the maximal concentration at each location.” I think I’m missing something. I don’t understand the need for a threshold if then an argmax is applied. Maybe clarify this part.

The used F-score (computation described in Appendix A) seems to boost performance of over-segmented predictions? To discuss.

How many WSIs are used for the Evaluation with Ground Truth Concentration Maps ? Is it described somewhere?

An analysis and discussion of failures could be interesting.

The conclusion could better discuss the limitations, the future work, the real clinical benefit, and the potential use of such methods in other applications.

“micorenvironment”->microenvironment

“with it’s absorption”-> its


**Paper Type:**

methodological development

**Questions To Address In The Rebuttal:**

The paper is of good quality in my opinion. The authors should fix minor typos, clarify a few points as mentioned in the comments above, and better discuss the results, limitations and potential future works.

---

### Official Review · Reviewer_4AxV · 2023-02-05

**Confidence:** 4
**Preliminary Rating:** 4
**Recommendation:** Poster

**Summary:**

The paper proposes a unsupervised method for the identification of different types of stains in Multiplex Immunohistochemistry (mIHC) images. Additionally, a inversion regulation technique has been proposed to handle the underdetermined nature of the formulation. The experiments have been performed on the single IHC image dataset for the analysis of the proposed methodology.

**Strengths:**

The paper is easy to follow, and presented comprehensively. The analysis presented is also extensive, and covers the different aspects of the methodology, for example ablation study presented in Table-3 ( appendix).  The approach to add the MSE(Y_inverse, S_inverse · C).
as the regularisation is also logical. Compared to the supervised counterparts, the approach is performing in the same vicinity.

**Weaknesses:**

The presented results are over a single iteration. These should be presented as a mean and standard value over multiple iteration (say 5). The qualitative result comparison of different methods is not included.  The impact of λ1 and λ2 in Equation 6 is not analysed. What is the sensitivity of the methodology to the hyper parameters? It would also be interesting to analyze the impact of network (backbone) choice on the performance. However, it is not covered in the results. For example, what if U-Net is used instead of vanilla autoencoder? The authors should also discuss why there is large variations in performance for different decomposition (CD3, CD4..). Results on additional datasets should be included.

**Deanonymize Review:**

no

**Detailed Comments:**

The captions of the figures and table should be more detailed.

**Paper Type:**

methodological development

**Questions To Address In The Rebuttal:**

Please refer to the weaknesses section for details. Additionally,
1. For Table-2 other similarity metrics should also be included. 2. What is the performance of the vanilla autonencoder (present backbone) supervised performance?

---

### Official Review · Reviewer_N8RJ · 2023-02-08

**Confidence:** 4
**Preliminary Rating:** 4
**Recommendation:** Poster

**Summary:**

In this paper, the authors introduce InverseAE, a method for unsupervised deep learning stain decomposition. At each pixel, they assume there is a dominant stain that should be attributed to that pixel in the final stain segmentation maps. Reconstruction component based on AE and regularization component based on inverse reconstruction comprise the proposed method.

**Strengths:**

- The proposed method only requires the reference stain colors. And this method greatly solves the previous method's dependence on expensive labeled mIHC data, and helps to improve the application of deep learning-based methods in this medical scenario.
- The regularization strategy based on inverse reconstruction is novel and well-suited to the task.
- The paper is easy to follow and well-motivated. Especially, the technique description section illustrates the role of each component clearly.

**Weaknesses:**

- In the table comparing method performance, the network structure and parameter size, which typically have a significant impact on performance, are missing. Additionally, it would be better to compare the training time.
- Using more image quality metrics might help to better illustrate the superiority of InverseAE. These commonly used metrics include MSE, PSNR, and LPIPS.

**Deanonymize Review:**

no

**Paper Type:**

methodological development

**Questions To Address In The Rebuttal:**

- Will the performance improve if replacing the convolutional AE with UNet-like structure?
-  Are the parameters and training time of InverseAE much higher than that of colorAE?
- How does the image generated by InverseAE perform on other image quality indicators?

---

### Meta-Review · Area_Chair_RWSd · 2023-02-25

**Recommendation:** Accept (Poster)
**Confidence:** 4

**Metareview:**

Strengths: Reviewers found this approach to be novel and interesting, and that it improves upon prior methods that require expensive, labeled data. The proposed regularization strategy is well-motivated. The writing was clear, and experiments/analysis were sufficient. Further quantification was performed based on the main review comments, including experiments with network architecture.

Weaknesses: it was noted that while the method is not highly innovative, the advance is solid and no major weaknesses were found. Evaluation on additional datasets would strengthen the results, though the availability of more data with multiple biomarkers may currently be limited.